# Brazilian Circular Economy Pilot Project: Integrating Local Stakeholders' Perception and Social Context in Industrial Symbiosis Analyses

Emilia Faria [1,*], Cristiane Barreto [2], Armando Caldeira-Pires [2], Jorge Alfredo Cerqueira Streit [3] and Patricia Guarnieri [3]

1   Sustainable Development Center, University of Brasília, Brasília 70910-900, Brazil
2   Mechanical Engineering Department, University of Brasília, Brasília 70910-900, Brazil
3   Post-Graduate Programme in Management, University of Brasília, Brasília 70910-900, Brazil
*   Correspondence: emiliaofaria@gmail.com

**Abstract:** This paper aims to analyze organisations' behaviour in the Industrial Symbiosis implementation process in the Circular Economy Pilot Project, in Brazil from the actors' perception. We conducted an exploratory and descriptive study with a qualitative approach to attain the research objective. The data collection involved in depth interviews with eighteen actors. Data were analyzed using the Content Analysis technique. The study results in show a still incipient industrial symbiosis network, with few connections between industrial actors, based on bilateral exchanges of materials, water and energy. From the analysis, it was possible to identify the elements that influence the behaviour of organisations. When it comes to exogenous elements, it is clear that laws and sanctions are the ones that most determine organisational action. However, this pressure is still focused on the traditional and unidirectional model of production processes. Economic viability was identified as a primary factor for the objective elements of organisational action. Regarding the subjective elements, it was found that there was already an interaction before the project between some companies, especially those from automotive sector. With the project's initiative, this social interaction was intensified, including between companies from different sectors. Regarding the barriers, we found the absence of governmental actions, unavailability of time and involvement of the managers to fully participate in the project, high cost of waste disposal versus the cost of investment and, discontinuity of actions. Overall, the study indicates that the project increased the institutional capacity of the region to develop industrial symbiosis, as it advanced in sharing new knowledge, promoted more significant interaction between organisations and identified business opportunities for companies. However, it appears that the project's continuity will be conditioned to improve some aspects of the governance structure, regulatory framework and collective engagement. These results can be helpful for researchers studying this topic and managers in Brazil and other emerging countries in Latin America, as well as, policymakers involved in public policies aimed to enable the transition to a circular e more sustainable model.

**Keywords:** industrial symbiosis; industrial ecology; circular economy; Brazil

## 1. Introduction

Despite the industrial sector's contribution to world economic development, it is known that it is responsible for much of the environmental degradation and scarcity of natural resources [1,2]. In a description of the current state of the raw materials recovery sector in Poland, a dynamic development of the recovery industry was observed, followed by formation of new companies, increasing amount of people employed (similar to 70,700 people), and a growing level of processing of secondary raw materials with the use of more and more innovative technologies [3].

The Industrial Ecology (IE) literature emphasizes the role of companies as a necessary component of change because of their potential for environmental improvement. This perspective arises with the proposal of a new model of industrial development in which it is possible to harmonize production processes and the limits of nature [1].

In recent decades, advances related to the inclusion of s environmental issue in organisations' guidelines, leading to changes in their internal processes and the way they relate to the environment [2]. In this sense, a part of IE, known as Industrial Symbiosis (IS), proposes a new organisational arrangement in which different industrial actors, from their underutilized resources, connect their flows. The secondary materials, water, energy resources, services, infrastructure and technology are revalued and reutilized to gain competitive advantages and reduce the environmental impact [4–6].

Applications of this model can be found in different parts of the world. At a regional level, the NISP (National Industrial Symbiosis Programme), the United Kingdom's IS program, presents essential data related to the reduction in the amount of waste dumped in landfills, the use of virgin raw materials and the emission of $CO_2$. In addition to the decrease in disposal, storage and transport costs and revenue generated by additional sales [7].

At the national level, countries such as China [8], South Korea [9] and Japan [10] develop programs based on the principles of IE and Circular Economy (CE). At the local level, initiatives to transform traditional industrial districts into Eco-industrial parks (EIP) stand out, such as the By-Product Synergy (BPS) project in Altamira, Mexico [11], cities from Kwinana and Gladstone in Australia [12] and Kalundborg, in Denmark [13].

As observed by Rincón-Moreno et al. (2020) [14] and Neves et al. (2020) [15], most of the case studies reported in the IS literature came from China and countries in North and North-West Europe. It clearly shows a literature gap related to investigation reports from other continents. Related to emerging countries, there are two hypotheses, IS is still not widespread, or such phenomena have not been recognized or described yet [16].

Malaysia is an example of a developing country whose majority practice is still the end-of-pipe approach, which promotes landfill disposal methods. Many companies do not have professionals specialized in solid waste management because this is not a significant concern [17]. Data from the Brazilian reality corroborate this understanding. From 2010 to 2019, Brazil's urban solid waste generation increased from 67 million to 79 million tons per year. Even though the amount of waste collected has grown from 59 million tons in 2010 to 72.7 million tons in 2019, most of them are still disposed of in landfills and inappropriate units such as dumps [18].

Mapping the Brazilian literature, few studies were found in this area, especially when dealing with already implemented IS cases [19]. Two initiatives stand out, the Rio Ecopolo Program [20] and the Minas Gerais Industrial Symbiosis Program (PMSI in Portuguese) [21]. The first, inspired by the international experiences of EIP, was conceived as an environmental planning strategy to promote sustainable development and improve urban and environmental conditions in the metropolitan region of Rio de Janeiro/RJ. However, political changes meant that the program did not move forward.

On the other hand, the PMSI, an initiative of the Federation of Industries of the State of Minas Gerais (FIEMG in Portuguese), managed to obtain good results from the IS concepts in the state of Minas Gerais [21]. Because of the successes achieved, FIEMG restructured the PMSI, changing its locus of action to industrial districts and expanding its scope to the CE [22]. The industrial district of Sete Lagoas was chosen to receive the program's pilot project in 2017, becoming one of the only Brazilian initiatives in the IS implementation phase at the district level.

More recently, the concept of CE has been unified with the theoretical body of IE and has gained notoriety. CE is more present in policies, business guidelines and academia as a model capable of replacing the unidirectional and traditional logic of waste disposal. CE aims to create a model of circularity and valorization of resources. From the CE's

perspective, IS is considered a sustainable business model based on innovation and collaboration [23,24].

Despite the CE and IS relevance, there is a lack of empirical studies on the role of local stakeholders, such as individual firms working in collaborative groups, in developing a local CE [14,25]. In this same logic, it is also necessary to investigate how common beliefs, values, and norms develop within a social structure and how these influence the level of engagement in collaborative projects such as IS [26].

Recent initiatives provide assessments of IS based on contextual factors. Mainar-Toledo et al. (2022) [27] advocate that more than technical and economic attractiveness and an enabling legal and policy context is needed. For cooperation solutions, historical, cultural, organizational, social, and behavioural factors also play a significant role.

In order to, fill this gap, [20–22] this study intends to extrapolate the technical and economic aspects of IS, integrating the perception of local stakeholders and the social context in which these interactions occur in the analyses. Its relevance is justified as it investigates an industrial development model capable of transforming current business practices. By connecting different industrial actors through mutually beneficial transactions from an environmental, economic, and social standpoint, IS proposes creating a new organizational culture founded on cooperation, collaboration, and trust.

Given the scenario of scarce literature on the subject in Brazil and few initiatives implemented, the overall objectives of this case study were to identify the evolutionary trajectory of IS in the industrial district of Sete Lagoas and analyze the behaviour of organisations in the IS implementation process in the CEEC Pilot Project of the industrial district of Sete Lagoas from the perception of the main actors involved.

## 2. Theoretical Background

### 2.1. Industrial Symbiosis

The EI study field emerged when the environmental impacts of human activity were little discussed, and corporate and government initiatives related to environmental preservation needed to be more coordinated [28]. In a seminal study on IE, Frosch e Gallopoulos (1989) [29] already emphasized the need to transform the traditional industrial economic system into an integrated model in which matter conservation processes were a priority.

The emergence of the field pointed to a real need to understand better the complex links between industrial systems, human society, and the biosphere based on a new model that was analogous to the biological ecosystem due to its characteristics of integration and cycle of matter [1,28]. The field focuses on preserving the ecological viability of natural systems, ensuring people's acceptable quality of life, and maintaining the economic viability of commercial and industrial systems [30].

At the inter-firm level, IS, a subfield of EI, emerged as a cooperative structure for exchanging materials, water, and energy between different organizational units. The essence of IS lies in the involvement of traditionally separate organisations in a network of synergies to promote eco-innovation of their processes and products and long-term cultural change [5,31]. Therefore, in this study, IS is defined as a complex social process in which different industrial actors trade their underutilized resources. These trades can be made through (1) the use of by-products, water and energy and/or (2) the sharing of services, such as the use of collective infrastructure, logistical services and environmental activities [4].

This interaction between actors in mutually beneficial transactions from an economic and environmental point of view can trigger the mobilization of intangible assets, such as intellectual and social capital, and create a collaborative culture. Within this concept, several arrangements exist for the materialization of IS, such as EIP, virtual IS networks and industrial ecosystems [4].

Although the initial development of IE was based almost exclusively on technology-based arguments [1], there has been a movement that values the contributions of the social sciences to the field. This approach seeks to understand to what extent material

and energy flows are shaped by the social context in which they are inserted [32]. In this sense, studies have been conducted emphasizing the interactions of the industrial system with the environment. Contextual elements, organisational structure, interaction patterns, managers' beliefs and how governments try to influence the behaviour of organisations began to compose the analyses.

The UK has several IS initiatives fostered by the NISP. In the first studies, [33] high-lighted the relevance of some factors, such as the nature of the companies' operations and industrial history, peer pressure and the coordination mechanism. In other analyses, the factors of geographic proximity and social interactions were questioned by [34] and [31].

In Denmark, the IS reference case in Kalundborg brings several contributions to the literature. Authors report that the emergence of the first interactions between companies was due to the great water deficit in the region [35,36]. As it is an industrial district with a diversity of sectors with different inputs and outputs, companies saw possibilities to develop symbiotic relationships [6,30,37]. Based on the bilateral agreements, the economic viability of the exchanges was verified [38,39]. However, only these variables could not explain this change in the behaviour of organisations. Studies have advanced towards understanding the social context of these relationships.

In these cases, legislation and the relationship between government and industry were also important during the process. Although there was government intervention through command-and-control instruments, the Danish regulatory system advocated a voluntary and more proactive approach by companies [30,35,37,38].

In line with the Kalundborg case, the Kwinana IS in Australia also arose from a concern about water scarcity [40,41]. In order to solve this issue collectively, an industry council was created. With the establishment of this organisation, communication and interaction between companies were facilitated [41]. Companies in the same segment constantly shared information intending to improve the efficiency and performance of operations.

Ulsan, South Korea, is another case of an industrial district that has changed its practices because of its impact on the environment. Due to stringent environmental regulations imposed by the government, industries have had to adapt by investing in pollution prevention equipment, establishing cleaner production practices and implementing ISO 14001-based environmental management systems [42]. Additionally, the government of South Korea has established a national IS program with a local governance structure capable of mobilizing business actors, universities, research institutes and local governments in the transition process [43].

Vanhamaki et al. (2020) [44], in a study realized in Finland, ponder that regulations need to support the implementation of effective symbioses emerging from new solutions but also safeguard the environment and human health when closing biological loops.

Other factors have also been reported in the international literature. In a comparative study between US and Dutch Eco-industrial park, Dutch Eco-industrial park was more successful because it emerged from integrated initiatives between companies and local government [45]. In another study, in which the key factors for the viability of two IS projects were evaluated, one in the United States and the other in Ukraine, the champion figure, trust, community participation and geographic proximity were the most relevant [46].

In Puerto Rico, technical factors such as low volume of discarded materials, incompatible by-products between industries, an abundance of water and good landfill capacity did not create incentives for the reuse of materials. As for social factors, the lack of communication and little interaction inhibited the construction of a shared identity and the expansion and maintenance of synergies [32].

More recently, three drivers were identified from IS experience in Tanzania's largest integrated sugar refinery. First, regulatory pressure-the constant monitoring by the enforcement of water abstraction and wastewater discharge, has encouraged investments in by-product resource utilization strategies. The regulation boosted strategic responses, including water efficiency and recycling innovations. Second, the parent company's culture of continuous improvement-the culture of assessing opportunities for constant improve-

ment in value creation and business growth had significant impacts related to advances in the co-generation and irrigation technologies. Third, the inter-subsidiary competition-the competitive pressures combined with the strategic flexibility resulted in more investment in efficient, productive processes [47].

Rincón-Moreno et al. (2020) [14] bring a new perspective to IS studies by studying SMEs (Small and Medium-sized Enterprises) challenges in developing a CE system. Despite the possibilities of synergies pointed out and the interest of some companies in improving some aspects, such as sharing the infrastructure and carrying out joint activities, there needs to be more inter-organisational collaboration and a misunderstanding of the roles played by SMEs in developing the CE system.

In the context of developing countries, Noori et al. (2020) [48] bring the concept of an Emerging Industrial Cluster (EIC), a cluster in the first stages of development while expected to expand rapidly. Allied with IS practices, EICs can play a significant role in the industrialization of these countries. The Persian Gulf Mining and Metal Industries Special Economic Zone in Iran were used as a case study. Based on the social forces that shape the emergence of IS [4], the results of the study pointed to the relevance of these factors: infrastructure readiness, governmental financial stimulation policies, anchoring, and government planning dynamics, available information, monitoring and environmental assessment by governmental organizations and short or long-term business opportunities.

The park's transition towards industrial symbiosis and resource sharing was also analyzed in Taiwan. The transition was possible because of coordinated government support, technological subsidies, policy support, and willing manufacturers [49].

A qualitative multiple-case study of two Finnish IS cases examined its processes regarding their actors (public, private), IS level (person, organization, network), and IS process phases. As a result, the authors highlighted the relevance of the interplay of intertwining public and private agencies within each process phase and the engagement of several actors on different IS levels [25].

In the same logic, an analytical approach was applied in Tianjin Economic-Technological Development Area (TEDA), China, to determine the factors that significantly influence companies' participation or not in IS. For participant companies, the requirement of environmental regulations was the first driving force, and the uncertainty of waste and by-products is a paramount concern. Meanwhile, for non-participants, geographical disadvantages are vital restraints [50].

### 2.2. Theoretical and Analytical Framework

The foundation of this analysis is based on the theoretical understanding of the Institutional Analysis and Development (IAD) framework developed by Vincent and Elinor Ostrom and other scholars affiliated with the Workshop in Political Theory and Policy Analysis they established at Indiana University in 1973. The IAD framework has been used for over three decades in multiple disciplines, such as Economics, Political Science, and Sociology. It aims to understand how institutions operate and change over time [51]. The framework can be used to analyze static and dynamic situations where individuals develop new rules, new norms, and new technologies. As a diagnostic tool, it can be applied in investigations in which humans repeatedly interact in a context of rules and standards that guide their choices of strategies and behaviours [52]. In addition, Ostrom's framework also allows the incorporation of different stakeholders in the analysis and design of policies. It gives the framework the potential to produce a deep understanding of social situations. In addition to providing the basis for more effective policies, this understanding can build consensus for coordinated actions [53].

The core components of the IAD framework are the action situation, an abstraction of the decision environment in which a set of actors interact to make decisions, engage in relationships, and realize outcomes from their interaction; the inputs, which are organized into three categories of contextual factors: attributes of the community, rules in use, biophysical conditions. The rules in use are related to all aspects of the institutional context. The

attributes of the community designate all relevant aspects of the social and cultural context as trust, reciprocity, shared understanding, and social capital. The biophysical conditions refer to all aspects of relevant physical constraints on action situations. Another component is the actors that bring their existing repertoire of decision processes and capabilities to any action situation. The last one is the outcomes shaped by both the outputs of the action situation and by exogenous factors [54].

Although the studies mentioned above point out the relevant factors for the IS process in different contexts, it was based on the theoretical-analytical framework proposed by Faria et al. (2021) [55] that a more comprehensive and refined understanding of the evolution of IS became possible. The proposed structure (Figure 1), based on the IAD framework, comprises a universal set of elements from the social, economic, institutional, and physical environment that shape the behaviour of organisations towards IS practices.

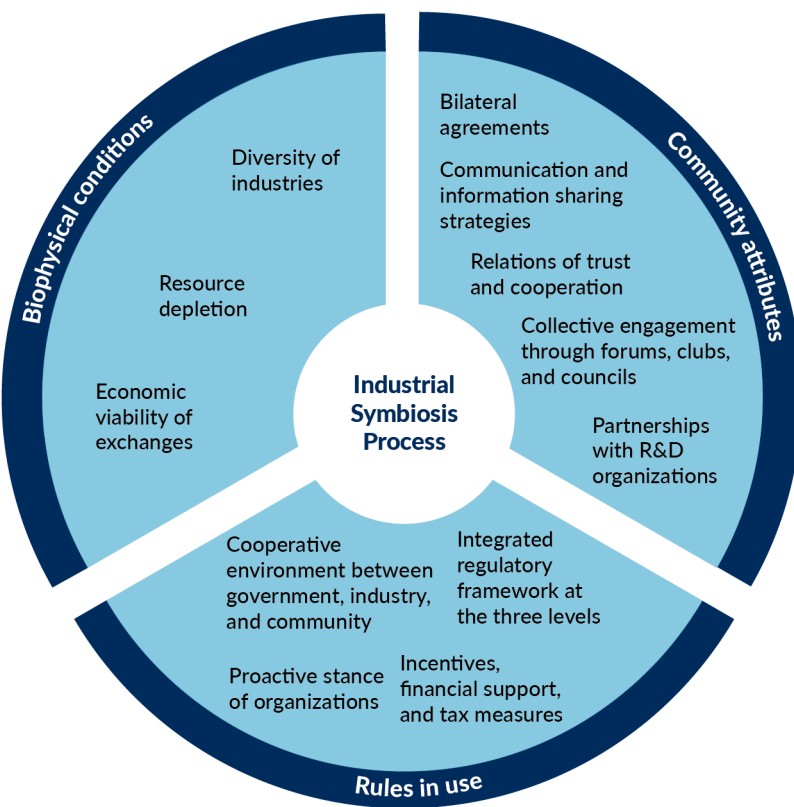

**Figure 1.** Theoretical and analytical framework for the IS process. Source: [55].

The pertinence of this framework is justified for three reasons: first, it considers the transaction costs associated with implementation policies [56]. Transaction costs are central to the study of economics as they explain the different organizational forms prevailing in markets. They refer to costs resulting from exchange processes between companies and can be divided into four types: (1) research costs, (2) contracting costs, (3) monitoring costs, and (4) execution costs. Research costs include gathering information to identify and evaluate potential business partners. Contracting costs refer to costs related to negotiating and writing a contract. Monitoring costs are associated with monitoring the contract to ensure that each party fulfils a predetermined set of obligations. Execution costs refer to the costs associated with negotiating ex-post and sanctioning a trading partner that does not perform according to the contract [57–59].

The second reason is the emphasis on contextual conditions (such as physical, social, economic, and cultural factors) influencing institutional design and performance [56]. In organizational theories and business practice, organizations were seen as autonomous and independent entities inserted in a particular environment that sought the best competitive

strategy based on their resources and internal capabilities [60,61]. However, several governance structures began to emerge to reduce these transaction costs, and new productive structures proliferated from the 1970s onwards. Thus, organizational approaches started to consider the organization from its insertion and interaction in a broader context [59,61,62]. Furthermore, the IAD contains no normative bias regarding the institutional arrangement used to implement policies. The framework does not establish the best arrangement to be used; it can be centralized, decentralized, or polycentric [56]. Finally, the IAD presents criteria to identify the strengths and weaknesses of different institutional arrangements, such as efficiency, equity, legitimacy, participation, accountability, adaptability, resilience, robustness, or sustainability [51,56].

*2.3. Industrial Symbiosis in Brazil*

In developing countries, the concepts of IS and IE are still not widespread and their practices are incipient [16]. However, implementing IS from an integral perspective allows these countries to develop industrially and socially without incurring the risks of depletion of natural resources [63]. CE, IE, and IS could be an essential means of establishing the re-industrialization of Brazil, with sustainability as the critical point of its strategic positioning [20].

According to Boom Cárcamo e Peñabaena-Niebles (2022) [63], Brazil is the country that has developed the most concerning the use of solid waste through IS in Latin America. Nevertheless, through a systematic literature review of the Brazilian scientific production of IS, Faria et al. (2022) [64] and Colpo et al. (2022) [65] showed that there remain few applications of the IS model compared to other countries.

Great part of the literature deal with the potential application of the IS principles and possible solutions for the Brazilian industry [65] Although solid waste management is a practice with advances in Brazil, there are few initiatives at the level of projects and programs (Table 1). One of them is the Rio Ecopolo program that has not evolved as expected. Changes in political administrations caused the state government to withdraw EIP support [20,66]. Another part of the literature addresses specific sectors such as metallurgy [67], steel and cement [68], furniture [69] and forestry sector [70].

**Table 1.** IS Programs and Projects in Brazil.

| Authors | Level | Main Contributions |
|---------|-------|--------------------|
| [66] | Rio Ecopolo Program | Features: Pioneer initiative in Brazil; systematic integration process among professionals in the environment area; associations of local industries as articulating agents in the exchange of information; periodic meetings for decision making; punctual and timid practices. Recommendations: conduct the process by the industrial representation entities instead of the government; greater recognition by industries of the importance of the project; government involvement through the incorporation of the theme into public policies, greater involvement of environmental agencies in inducing this type of program. |
| [20] | Rio Ecopolo Program | Context: collaboration between governments, private institutions and industries, communities and academia has not evolved as it should; changes in public administration caused the state government to withdraw support for the Program. Recommendations: a planning strategy for the state's sustainable development based on a concrete partnership between all the actors involved; community participation in the initial planning process to avoid misunderstandings, share responsibility and build trust; the government's role in disseminating and providing necessary legal support; community and industry education on these topics. |

**Table 1.** *Cont.*

| Authors | Level | Main Contributions |
| --- | --- | --- |
| [71] | Green City Project-Footwear Sector | Motivations: inspection and heavy fines motivated the adoption of new solutions related to waste; recognition of small businesses' inability to make changes individually led to a joint effort for a solution coordinated by the trade association.<br>Features: several meetings to convince companies to transform their practices; the leadership of the association's director facilitated building trust; collective learning through training and development of educational material; waste management infrastructure for monitoring information and providing reports to companies. |
| [21] | Minas Gerais Industrial Symbiosis Program | Features: coordination made by FIEMG; the sparse network with few waste exchanges; the effort of the actors involved in building knowledge, mobilization and relationship capacities; institutional context unfavourable to developing industrial symbiosis.<br>Recommendations: it needs to invest in eco-innovative solutions and improvements in the institutional environment to develop efficient waste management. |

Source: [64].

### 2.4. Case Study Context and Its History

Some IS projects emerged in the 1990s. One of them, the BPS project started in 1997 by the Business Council for Sustainable Development of the Gulf of Mexico, aimed to reduce pollution and save money and energy by reusing and recycling waste materials across factories, businesses and the local community [11]. This project served as inspiration for the first IS initiatives in the UK.

Based on the results achieved by the NISP, many countries were interested in replicating its methodology [72]. In a partnership between the United Kingdom and Brazil in 2009, FIEMG created the PMSI to promote profitable interactions between companies from all industry sectors [73].

FIEMG conceived the Circular Economy Program in Industrial Districts. Their objectives are disseminating the CE concept, the proposition of collective businesses aimed at reuse, and incorporating resources from the production process. In addition, reducing operational costs improves environmental indicators and attracts industries and investments to the region. Finally, CE aims to increase cooperation between local industries and the mining industry's competitiveness [74].

In Brazil, industrial concentrations arose spontaneously without the participation of planners in determining their location. Investors chose to install their industries in areas with more advantages, such as raw materials, labour, land, energy, water, transport facilities, etc. This way, industries were located near or in large urban centres, mainly in the Southeast Region of Brazil. The industrial districts were built to decongest and order the industrial expansion of large industrialized centres, encouraging the industry to locate in areas previously prepared and chosen following the development policy of each state. However, there was no national or regional policy for implementing industrial districts, and it was up to the states and municipalities to decide to build and manage them [75].

In Minas Gerais, a pioneer state in Brazil in implementing industrial districts, the activity gained strength in the 1970s when it sought to decentralize business investments, to develop and strengthen the municipalities in the interior. The attraction and installation of new industries generated a substantial economic impact, such as an increase in municipal revenue and the number of direct and indirect jobs. Currently, the state has 53 industrial districts [76]. Among them, FIEMG chose the industrial districts of Sete Lagoas to receive the first CE pilot project.

The industrial districts of Sete Lagoas (ID-I and ID-II) are located in the Central Region of the State of Minas Gerais, close to the Metropolitan Region of Belo Horizonte (capital of the department). These are two discontinuous areas, and Industrial District I, with a total

area of 1,512,230 m$^2$, was implemented in 1974 and Industrial District II, which occupies an entire space of 266,067 m$^2$, is currently being implemented. In 2016, the Action Plan for the Revitalization and Modernization of the Industrial District of Sete Lagoas was prepared. This plan was to organise new and modern industrial districts [77]. This study served as the basis for defining the location of the state's first CE pilot project.

## 3. Methodological Aspects

### 3.1. Methodological Choices

Qualitative research aims to explore human and social phenomena as they occur in their natural environments, based on understanding and interpreting phenomena in terms of the meanings people attribute to them [78,79]. Researchers who opt for a qualitative approach are usually interested in understanding the participants' life experiences in complex themes that cannot be represented in numbers [80].

Applied research was chosen to achieve the article's objective, with descriptive goals. Applied research because it collaborates with resolving a real problem (early stage of Circular Economy in Brazil). According to Patton (1990) [81], studies of this nature bring light to situations that still lack a practical resolution. The descriptive objectives are characterized by the intention of sharing the players' experiences, highlighting them in line with the research objectives [81].

As a technical procedure, documental analysis and in-depth interviews were used. Secondary data were obtained through documentary research that involved the analysis of official documents, reports, and electronic sites. We also consulted documents produced by the entities involved in the project implementation. In addition, to gain more knowledge on the subject, document analysis helps learn about examples implemented in other contexts and obtain updated statistical data. Finally, document analysis is vital for creating categories for subsequent data analysis [82].

From the documental analysis, the actors to be interviewed were identified. The CE pilot project was conceived and coordinated by FIEMG, supported by the Commercial and Industrial Association (ACI) of Sete Lagoas and had the participation of a local university centre. The project had 24 participating companies, of which one was no longer in operation, and of the remaining 23, all were contacted, and 15 were available to participate in the interviews. Thus, the criteria for choosing the interviewees were based on representativeness and accessibility.

The primary data, in turn, were collected through in-depth semi-structured interviews. Semi-structured interviews are based on a script. However, there is flexibility for the researcher to add or delete questions in the execution in order to obtain more coherent and sincere information [82]. The in-depth interview technique was used as it allowed for exploring the subject's perspectives on a given idea, program/situation or exploring new issues in depth [83]. Below, Table 2 summarizes the main methodological classifications of this research.

**Table 2.** Research methodological classification.

| Aspect | Classification | Reference |
|---|---|---|
| Approach | Qualitative | [78–80] |
| Nature | Applied | [81] |
| Goals | Descriptive | [81] |
| Data collection techniques | Documentary research and in-depth interview with a semi-structured script | [82,83] |
| Triangulation | Collection methods | [84] |
| Data analysis | Content analysis | [85,86] |

The following subsection will discuss the data collection and the instrument created for these.

### 3.2. Data Collection

Data collection used a combination of sources that included both primary and secondary data. In the social sciences, it can be called triangulation of data collection instruments when the researcher uses multiple ways to obtain data. Triangulation is recommended to bring more robustness to scientific research [84].

Two different scripts were built. One was aimed at business actors, and the another at institutional actors. Both were constructed from the structure of the IAD framework to help understand the phenomenon of study that involves several variables that affect human action in the choices of strategies and behaviours [52], which means that the categories of analysis were defined *a priori*. One of the objectives of the IAD is to identify the contextual factors that encompass aspects of the social, cultural, institutional and physical environment in a given situation. Therefore, the following categories were used action situation, biophysical conditions, rules in use and attributes of the IAD community in the elaboration of the script [51]. Additionally, the papers of Boons et al. (2014) [87] e Faria et al. (2021) [55] also support IS questions.

In total, 18 in-depth interviews were carried out, 15 respondents linked to the companies and three respondents from the institutions involved. Table 3 details the information of the 15 respondents and the companies to which they are linked. Interviewees' names were not used to preserve the identity of the people involved. For companies, fictitious names were created based on the economic activity developed.

**Table 3.** Datasheet of respondents and economic activities of companies.

| Interviewed | Post | Main Economic Activity | Fictitious Company Name | Date |
|---|---|---|---|---|
| ENT1 | Environmental analyst | Automotive component manufacturing | AutoComponent | 19 March 2021 |
| ENT3 | Majority Partner and Administrator | Manufacture of sinks and tanks | SinkPlant | 6 April 2021 |
| ENT4 | Industrial director | Manufacture of car audio components | AutoAudio | 8 March 2021 |
| ENT5 | Workplace Safety and Environment Coordinator | Automotive component manufacturing | AutoSolder | 16 March 2021 |
| ENT6 | Senior Environmental Analyst | Manufacture of trucks and buses | CarPlant | 9 March 2021 |
| ENT7 | Integrated Management System Analyst | Production of denims and fabrics, yarns and denim | TextilePlant | 25 February 2021 |
| ENT9 | Industrial director | Metal Machining | MetalPlant | 5 April 2021 |
| ENT10 | Owner and Chief Financial Officer | Manufacture of ice cream, other frozen food and industrial bakery | FrozenFoodPlant | 22 March 2021 |
| ENT8 | Safety and environment supervisor | Production of steel forgings | SteelPlant | 7 April 2021 |
| ENT13 | Director | Coffee roasting and grinding | CoffeePlant | 22 April 2021 |
| ENT15 | Owner | Manufacture of meat products | MeatPlant | 20 January 2021 |
| ENT14 | Environmental analyst | Manufacture of cleaning and polishing products | CleaningPlant | 20 April 2021 |
| ENT16 | Owner and Industrial Director | Manufacture of cement artifacts for use in construction | CementPlant | 31 March 2021 |
| ENT17 | Manager | Wholesale trade of metal waste and scrap | WastePlant | 15 March 2021 |
| ENT18 | Commercial director | Activities to support livestock and dairy and animal feed manufacturing | DairyCoop | 16 April 2021 |

In addition, we delivered informed consent to all participants, in which we explained that the anonymity of the participants would be respected and the data would be analysed

in an aggregated way, exclusively for research purposes. Besides that, we explained the study participants' rights, following the Council of Research Ethics of Brazil (Conep). It is also important to point out that this research was exempted from being submitted to the Ethical Committee of the University coordinating this study. It did not conduct experiments with humans or animals and did not involve any vulnerable population or sensitive topic, requiring just the opinions and perceptions of managers related to IS and CE concepts.

Tables 3 and 4 contain the technical data of the institutional actors involved in the project. Their identity has also been preserved.

**Table 4.** Datasheet of institutional actors.

| Interviewed | Post | Institution | Date |
|---|---|---|---|
| ENT2 | Environmental Analyst-Coordinator of the Circular Economy Program | Federation of State Industries | 21 January 2021 |
| ENT12 | Executive Superintendent | Commercial and Industrial Association | 21 December 2020 |
| ENT11 | Professor | University centre | 1 April 2021 |

Of the 18 interviews, only one was carried out in person in the city of Sete Lagoas/MG, the others were conducted remotely through online videoconferencing platforms such as Zoom meetings and Teams. All interviewees allowed recording the interviews. The recordings ranged from 25 min to 1 h and 23 min and were later transcribed. The following subsection describes the data analysis supported by the literature.

Like any study, this article has limitations. The COVID-19 pandemic period during data collection made technical visits and on-site observation impossible. Therefore, only one of the interviews was carried out in person. This interpretation may have influenced the researcher's and interviewee's approximation and interaction. Another methodological limitation related to collecting primary data is that the researchers chose only semi-structured interviews. Survey could complement the qualitative data obtained to understand the same problem from a statistical point of view while covering more respondents.

*3.3. Data Analysis*

For the treatment and analysis of the interview transcripts, the content analysis method was chosen, which comprises three basic steps: First, pre-analysis, which refers to the selection of material and the definition of procedures to be followed. Second, exploration of the material that comprises the systematization and categorization of data. Third, data processing and interpretation regarding the generation of inferences and investigation results [85].

After an initial floating reading and organizing of the material, the interviews were systematized and categorized. Fragmentation, separation, word count and the formation of analytical clusters were a resource of the Nvivo 11 software that helped diagnose trends, concordances, gaps and complementary excerpts from the different interviewees.

Table 5 presents the structure of the categories and subcategories of the study constructed *a priori* and *a posteriori*. Bardin (2016) [85] states that *a priori* categories are those elaborated from common sense or already published literature. So, this classification can be found in classic works like [52] and more recent ones like [55]. The framework is divided into three broad categories: exogenous factors, inter-organizational action and barriers to implementation. The subcategories help explain the main categories of analysis from the grouped recording units.

**Table 5.** Structure of Categories and Subcategories.

| Categories and Subcategories | No. Sources | No Encodings |
|---|---|---|
| Category 1. Exogenous factors | | |
| Strategic alignment | 6 | 12 |
| Standardization | 8 | 26 |
| Legal compliance | 18 | 82 |
| Category 2. Interorganisational action Objective elements | | |
| Flow of information | 14 | 42 |
| Industrial diversity | 12 | 19 |
| Resource flow | 13 | 36 |
| Economic viability | 17 | 53 |
| Subjective elements | | |
| Coordination mechanism | 15 | 35 |
| Community interface | 7 | 15 |
| Organisational interactions | 12 | 34 |
| Category 3. Barriers | | |
| Absence of government actions | 12 | 28 |
| Unavailability of time | 2 | 3 |
| Destination cost versus investment cost | 2 | 4 |
| Discontinuity of actions | 7 | 14 |

For the categorization, the criterion for selecting the recording units was semantic. The theme is the unit of meaning released from a text according to specific criteria related to the theory that serves as a guide for reading. The analysis involved discovering these cores of importance that make up each text fragment whose presence and frequency can mean something to the research [85]. All registration units were grouped around a theme and became part of the categorical axes of the investigation.

According to Vaismoradi, Turunen, and Bondas (2013) [86], this way of treating qualitative data is characterized as a deductive approach to content analysis. After all, an existing classification was used to investigate a new phenomenon. According to the same authors, this phase of comparing response clusters to generate categories and subcategories can be called data organization. After this phase, the "Reporting" stage begins, presenting the data by comparing them and, preferably, using creative ways of exposing them.

## 4. Results

### 4.1. Circular Economy Pilot Project

The industrial districts of Sete Lagoas were chosen for the execution of the project that took place between 2017 and 2018. For the execution of the project, partnerships were made with the Commercial and Industrial Association (ACI in Portuguese) and the University Centre of Sete Lagoas (UNIFEMM in Portuguese). The methodology adopted was based on three steps, as shown in Figure 2.

In step 1, FIEMG sought partnerships in the city to leverage the project and ACI, as the only entity representing industry and commerce, promptly accepted to participate. UNIFEMM was also invited to the awareness-raising phase. From these first meetings, students from the institution's engineering courses were selected to support the project by collecting data during company visits.

In step 2, the identification of resources was carried out through questionnaires that sought to collect data on inputs, water consumption, effluents, solid waste, emissions, products, utilities and infrastructure. With this information mapped and tabulated, FIEMG and its specialists began the analysis of resources and possible synergies and business opportunities. Despite the project's scope for EC, there was a greater emphasis on proposals that involved the IS, such as the incorporation of malt bagasse in the manufacture of animal

feed, the burning of wood chips to generate energy/heat, the reuse of untreated water for cooling and refrigeration, washing floors and equipment, the raw material [73].

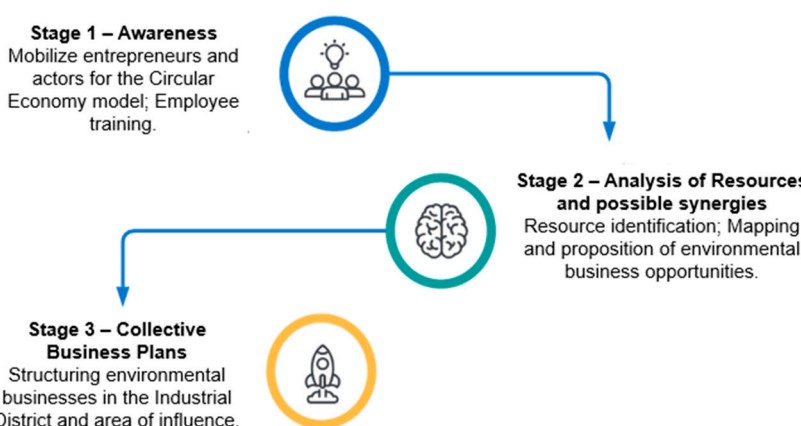

**Figure 2.** Project Implementation Methodology.

After the phase of proposing the plans, FIEMG was not involved in the formal aspects of the contracts, such as values, terms and conditions. This part of the negotiation was in charge of the companies. In other words, as soon as the proposals were presented, FIEMG ended its participation in the project, ending the methodology. As for the implementation and execution of the Collective Business Plans (PNCs in Portuguese), FIEMG did not follow the developments.

The information provided by FIEMG refers to the partial results of the project, such as the number of companies that joined the project, the description of each one, the number of CE actions mapped and the number of proposed PNCs.

### 4.2. Dynamics of Industrial Symbiosis in Sete Lagoas

The results presented so far have been restricted to describing the context of the creation of the CE project and the methodology applied by FIEMG to implement the IS based on documentary data and reports from institutional actors. However, to better understand the dynamics involved in the emergence and implementation of the IS, it was necessary to analyze the other actors' perceptions.

### 4.2.1. Category 1-Exogenous Elements

The way organisations behave is shaped by a set of social, cultural, economic and institutional conditions. Category 1, called exogenous elements, refers to factors the organisation partially controls or restricts its power to prevent. It is subdivided into three categories: Legal Compliance, Standardization and Strategic Alignment.

All speeches that addressed compliance with legislation cited the importance of the National Solid Waste Policy (PNRS in Portuguese). Therefore, waste disposal, inspection and licensing experiences were grouped around the Legal Compliance subcategory. According to Guarnieri et al. 2020 [88], this is the Brazilian law that comes closest to the objectives and principles of the Circular Economy. Therefore, the government of Brazil needs to monitor and enforce compliance with PNRS if it intends to move towards the CE.

The PNRS presents itself as the most representative subcategory within category 1, with 82 coded excerpts. It can be inferred from the predominance of this subcategory that the behaviour of organisations still has as its main component the fulfilment of legal demands. Therefore, companies comply with the legislation to avoid sanctions such as fines and suspension of operations. The speech below highlights this issue.

"The legislation that that we follow is national legislation. What I look for the most is in the composting area, because it is the area we work in. Therefore, I try to know what kind of waste I can use, what is the process, what is the legislation regarding what (ENT15)".

On the other hand, there was a gradual movement towards introducing environmental management and the theme of sustainability in the organisational strategy through adopting certifications such as ISO14001 and ISO 9001. These actions reported in the speeches mainly of the interviewees linked to the medium to the large size of the district constitute the Standardization subcategory. Seven of the fifteen companies interviewed are signatories of some ISO certification or are in the implementation process. The excerpts below bring these data, as shown in Table 6:

**Table 6.** Standardization Subcategory.

| Standard | Coded Excerpts |
| --- | --- |
| ISO 9001 (quality management system) | "We are environmentally certified, yet not in ISO 14000, not necessarily. We have a good part of the quality management system. We have a directive that deals with this. We have to have a well-publicized environmental policy within the company because our customers demand this from us, and most of them are certified (ENT9)". |
| ISO 14001 (environmental management system) | "( . . . ) Certification, we have 14,001. Today, AutoSolder works . . . I am talking about AutoSolder worldwide, because it is a Portuguese company, right? AutoSolder changed its management strategy a year ago where it includes sustainability as one of the company's pillars (ENT5)". |
| ISO 9001 and ISO 14001 under implementation | TextilePlant will also enter the certification part of 14001, so this question is essential for us. We have 9001 and are looking for 14001, which are already part of the Pirapora factory (ENT7)". |
| ISO 50001 (Energy Management) under implementation | "( . . . )We are there with an energy management system project. We are looking for certifications and an energy management system here at the company because the annual reductions are very significant for AutoComponent (ENT1)". |

In the last subcategory Strategic Alignment, the excerpts in which the interviewees referred to a practice adopted from the experience of other companies or to some change in the organisation's strategy were grouped. This subcategory was less expressive in the group of companies analysed, especially in smaller, more isolated companies, with little interaction with other companies and a view focused on costs. It was observed that companies linked to the automotive sector are more articulated with each other, promote joint actions, share experiences and consequently are influenced to incorporate certain behaviours. In the report by ENT4, he tells about the experience of the Pernambuco industrial park in which companies have a high level of synergy with frequent interaction between directors to discuss joint actions. From that case, he stated: "So, in light of this best case, we also started to bring these companies together to make the projects common, no matter how much they are in the electronics or stamping area, the projects are common, that they are replicated and used as good practices (ENT4)".

It is noticeable in the speeches of the interviewees linked to large companies this strategic alignment in which sustainability has become a pillar in recent years, no longer being seen only as a cost. The environmental issue starts to have a different focus. The environment's preservation becomes increasingly a requirement of the market, consumers, and trading partners. This is reflected in the speech of respondents. It is worth highlighting the discourse of interviewee number 8.

"Now, more than ever, they see that it is essential for the company to be aligned with preserving the environment because this makes their company more competitive. If today it is on the stock exchange, it shows that it is a company that thinks about society, feels about the environment, that thinks about the return for shareholders (ENT8)".

Another example that fits into this subcategory is an innovative compressor water reuse practice the Ent 1 company uses. Several other companies adopted this practice.

"Based on our project, the steel company also started to reuse this water. The group's companies now all adopt the same procedure. They managed to reuse this water in other companies. The project is already in four different companies (ENT1)".

4.2.2. Category 2-Interorganisational Action

Category 1 (Exogenous Elements) deals with organisational behaviour dynamics macro context. In addition, its subcategories have enabled a better understanding of the environmental conditions that influence the decision-making process of organisations towards more sustainable policies.

To broaden the understanding of the IS process under development in Sete Lagoas, exploring the meso-level elements that permeate the relationships between organisations becomes relevant. For this, category 2 Interorganisational Action was divided into objective elements and subjective elements.

The subcategory Objective Elements of inter-organisational action grouped the technical and economic variables that permeate the IS process. The variable "Economic Viability" is the strongest among the elements that explain the IS. Respondents understand symbiotic exchanges as an opportunity to increase revenue and reduce costs. In Table 7, excerpts are presented in which the interviewees explain the results of these actions from an economic point of view.

**Table 7.** Subcategory Economic Feasibility.

| Result | Coded Excerpts |
| --- | --- |
| Increase in revenue | "From a financial point of view, the company obtains resources from the sale of this wood (ENT1)". |
| Cost reduction | "The company that collected and brought us to the destination charged much more regarding the transport issue. So we did have the reduction. I can't tell you in percentage at the moment. But we did have a reduction in the value (ENT10)". |
| Increase in Revenue | "We invested in constructing a shed and yard to receive the waste. Thus, the production of organic compost increased, with no increase in costs for purchasing products. The opposite happened. We charge the reception of these products, which ended up generating revenue for the company (ENT15)". |

One of the main technical characteristics in the literature for developing IS is the need for different industrial types [89]. This "Industrial Diversity" element of the industrial district of Sete Lagoas can be verified from the following excerpt:

"( . . . ) Sete Lagoas is a city with diversified industrial characteristics, and here we have a large steel mill. Our park is not the largest in Brazil. It is one of the largest there. We have a textile industry, food factories, food factories, DairyPlant as the main one, and many others, and we also have a large automotive and industrial park in addition to Ambev (ENT4)".

The flow of information is another technical element to be considered in the interaction process between organisations. After all, it can be an obstacle to realising synergies or a dynamo to promote connections [90]. This element proved relevant during stage 2 of the project for identifying resources and mapping and proposing environmental business opportunities. All interviewees reported that the project team made technical visits to companies and shared information on the inputs and outputs of their production processes.

From the survey of the different industrial actors in a region and the sharing of information, it is possible to establish the "Resource Flows" that can be transacted. These connections can occur through (1) the use of by-products, water and energy or (2) the sharing of services and utilities, such as the collective use of infrastructure, logistical services and environmental activities [4]. The resource flows in the Sete Lagoas district were mapped from the interviewees' reports and are represented in Figure 3.

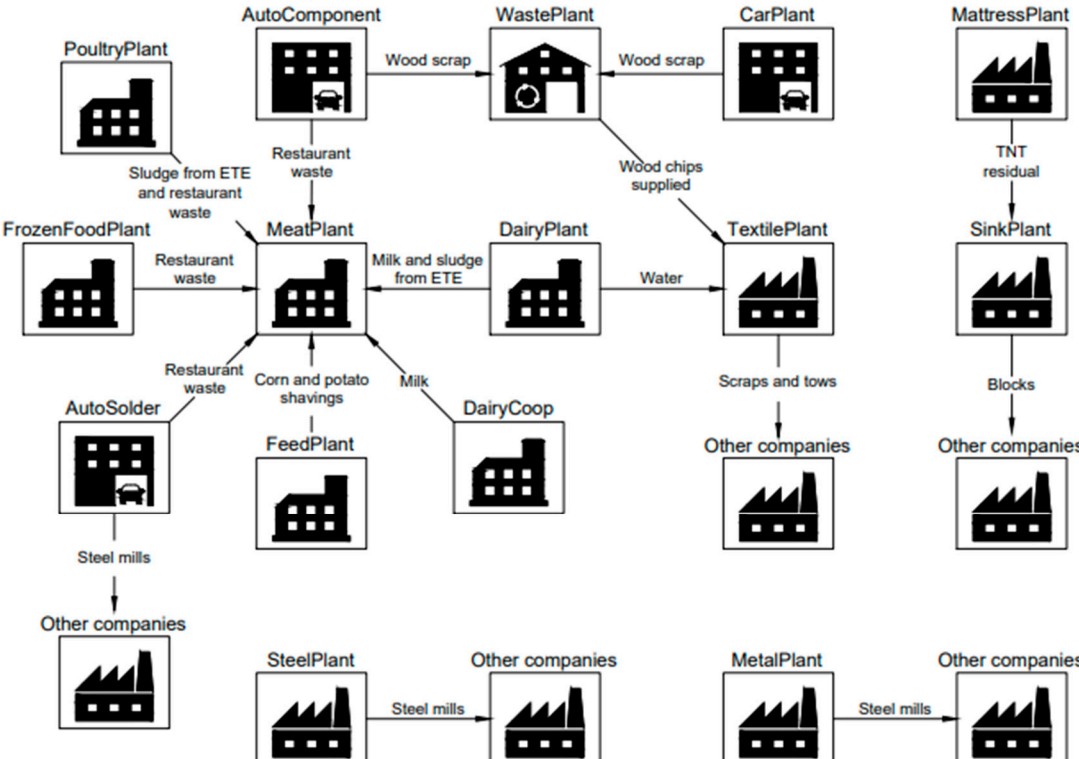

**Figure 3.** Flows of resources between the companies participating in the project.

MeatPlant works as a central point in the IS network of Sete Lagoas, containing seven interactions with different companies. The company receives restaurant waste from four industries: AutoSolder, FrozenFoodPlant, PoultryPlant and AutoComponent. From food companies, the farm gets corn and potato shavings, milk and sludge from sewage treatment stations (ETE in Portuguese). Of the waste received, solid waste has two destinations: animal feed and compost. The organic compost produced is sold and generates revenue for the farm. Liquid waste, such as treated effluents and milk, are applied in fertilisation.

Another essential agent in implementing the IS is WastePlant: a company specialising in industrial waste management. It is responsible for receiving large volumes of wood scrap generated by the AutoComponent and CarPlant industries. This wood scrap is crushed and transformed into chips supplied to produce energy for TextilePlant in its boilers. TextilePlant also reuses the treated effluent from DairyPlant in the fabric production process, which requires a lot of water. This flow occurs through a pipeline from one industry to another. The scraps and tows resulting from its production process are sold to small regional companies.

Industries in the metal and steel sector, such as SteelPlant, MetalPlant and AutoSolder, also sell their by-products to other companies, such as steel mills. Finally, the last flow is the commercialization of residual TNT from the production of MattressPlant to SinkPlant, which uses it to transport the parts produced. Residual pebbles from SinkPlant production are sold to block and moulded parts companies.

In summary, despite the efforts of FIEMG to propose the PNCs, it is clear that the CE project resulted in an IS network still in its infancy. After all, this CE project has just a few connections between industrial actors based on bilateral exchanges of materials, water and energy. As for the sharing of services, companies in the automotive sector reported that they have an integrated system of logistics services. However, it is not an initiative resulting from the FIEMG project.

As important as identifying the connections established by the flows of IS matter and energy between companies is understanding how they relate to the environment in which they operate. Therefore, the subcategory "Subjective Elements" covers the relationship

between companies and the community, interactions between organisations and the role of the champion in the Sete Lagoas DI.

The subjective element, "Interface with the Community", is intended to understand companies' engagement with the local community. The excerpts that reference the projects and actions that the companies carry out were grouped and listed in Table 8. It can be seen that some activities are by demand of the community itself, others in partnership with the city hall or other companies.

**Table 8.** Subcategory-Community Interface.

| Company | Social Actions |
|---|---|
| AutoComponent | • Improvements in the neighbourhood, such as a river catchment network, water reuse projects and a vegetable garden with composting in schools;<br>• Donation of pet tape and disposable material to an association of collectors. They return the material in the form of a pallet to the company;<br>• The reforestation project is in partnership with the city hall. |
| FrozenFoodPlant | • Cardboard and plastic donation to waste pickers association. |
| CleaningPlant | • Donation of PPE waste to a company that does the renovation and sells it at cost price to family businesses;<br>• Partnership with the city hall in the distribution of seedlings and, population awareness. |
| WastePlant | • Ponto Verde Project-individuals dispose of their waste at this collection point, and the company pays for it. |
| SinkPlant | • Donation of crushed stone waste for the community to use in constructions. |
| AutoAudio | • Donation of cardboard and a large volume of wood to the city's collectors association;<br>• Square revitalization by community demand. |
| AutoSolder | • Donation of wood, plastic, and paper to recycling company;<br>• Shared Solidarity Project-donation of food to the community in partnership with company employees;<br>• Cinema for children and actions on commemorative dates;<br>• Maintainer of the institution Next Step, which offers recycling workshops, seedling development;<br>• Plastic Fishing Project–cleaning the edge of the city's ponds in partnership with other companies. |
| TextilePlant | • Partnership with the environment department for the donation of drums;<br>• Participation in school events and commemorative dates. |
| MetalPlant | • Partnership with the city hall to install an ecological bus stop using pallets;<br>• Community garden inside the school and courses for the community. |

On the other hand, the "Organisational Interactions" element encompasses any type of relationship between two or more companies to achieve a common objective. This element proved to be more significant in medium and large companies. Before the CE project (Table 9), it was already possible to perceive some interactions, mainly between companies in the automotive sector. The interviews also identified other partners, such as the Brazilian Agricultural Research Corporation (Embrapa in Portuguese), the Agency for Technical Assistance, Rural Extension and Agricultural Research (Emater in Portuguese), and the University of Viçosa. These partnerships demonstrate the potential for companies to integrate with research institutes and universities.

**Table 9.** Organisational Interactions before the CE Pilot Project.

| Organisational Interactions before the CE Pilot Project |
|---|
| "AutoComponent is within a condominium of companies. In addition to AutoComponent there are four other companies within this same condominium. They are different companies with different segments, we have interaction here inside, and they have interaction with the companies that are around, which is the case of Maxion Montich, SteelPlant, and AutoSolder, then ends up having this interaction (ENT1)". |
| "We had few partner companies anyway. Because we produced a lot of compost on our own. We received from a single company (ENT15). |
| In general, we have a lot of relationships with this automotive, industrial park, and our relationship with AutoAudio is very much with this group. We have for another, and no longer for me as a director, but other positions such as HR, for example, which is in greater diversity, that there is a group that is us, there are other companies, CleaningPlant, TextilePlant, DairyPlant, there is a greater multidisciplinarity, but from the management point of view he is much more focused on automotive (ENT4)". |
| "( . . . )We attended 20 workshops from other companies to understand what they were doing. Of course, some we could not bring as process solutions, but many we managed to do just as we received seven or eight companies from groups of different companies. We showed the processes and projects they were trying, so this synergy has become very interesting. And we industrialists have been exchanging more stickers, which is very important. Our involvement with our primary client, CarPlant, has significantly helped in this cycle. We distance ourselves less, get closer to this operation, and want things to happen more efficiently, less painfully (ENT4)". |
| "( . . . )We deliver parts to them via a shared system. So, it means that the logistical efficiency, in this case, is 100%. They collect our loads. They collect loads from AutoSolder, AutoComponent, go to Betim, pick them up from Betim, go to São Paulo and deliver them to the automaker. And it works well, so it works. Obviously, we had to adapt so that we had greater liberality from the point of view of the control (ENT4)". |
| "We have more relationships with the security personnel, now within the pandemic, it stopped, but we already had monthly meetings each one in each company, presented something from the area that turned out to be relevant and other companies could use (ENT8)". |

The CE Pilot Project was an opportunity to strengthen these relationships. During the project, these interactions intensified, as confirmed by the following reports in Table 10. ENT13 points out that participation in the project was a chance for small companies to access larger companies.

**Table 10.** Organisational Interactions after the CE Pilot Project.

| Organisational Interactions after the CE Pilot Project |
|---|
| "From the meetings, I could visit CarPlant to understand the environmental issue. They also came here to the company to learn about environmental management. I was at AMBEV, a different company from the automotive sector, but we managed to exchange knowledge. We visited to learn about their management process. They were here to understand our process. So, it was cool. The companies were very participative. I still have contact with some, even though it happened in 2018 (ENT1)". |
| "With the circular economy project, which we started, for example, receiving cafeteria waste, which was an issue that we did not receive before. So, that was the big impulse after the project (ENT15)". |
| "But I talked with many. I remember talking with TextilePlant, I remember talking with, I think with MeatPlant, I believe these are the ones, but there were other conversations also between the companies that I remember like this (ENT16)". |
| "MeatPlant, through the circular economy program, we started negotiations to be able to recycle our organic waste, which is the waste left over from the restaurant, from the pruning of our gardens (ENT11)". |
| "So, participation within the circular economy opens up a range of visibility, even to know other companies inside and outside the state that may use these services in their process. This is very interesting, not only for this waste that we already know the destination within the circular economy. We can meet new partners, destinations or even waste that we can use as biomass in our boiler (ENT7)". |

"Do you know what generates? An approach of companies in several ways. For example, we do not have such great access to these multinationals. Here in Sete Lagoas, there are several, if you are part of these projects, sometimes you create a relationship ( . . . ) sometimes there is a solution for a multinational that you are looking for, and it is here inside the city".

Another variable that proved relevant for the IS process was the existence of local coordination or the support of a person with specific technical knowledge to conduct the process. This figure is called "a champion" in the literature, which can be a local leader, municipal authority, or class entity that promotes and advises IS actions. This subcategory was named the coordination mechanism.

There is a consensus among the interviewees that the ACI and FIEMG entities played this role in raising awareness among companies, organizing efforts, providing technical support and, above all, building trusting relationships. Table 11 describes these functions and the respective sections.

**Table 11.** Coordination Mechanism.

| Role | Coded Excerpts |
| --- | --- |
| Look for connections between actors | "ENT 2, I saw that he didn't get discouraged at any time and is still believing in the project and looking for connections. Seeking to favour sometimes in connection with the companies here, with companies in the neighbourhood. So, I see him as the central figure (ENT11)". |
| Mobilize all stakeholders to participate in the project | "When FIEMG brought the project here in Sete Lagoas, they sent an invitation letter to all companies. We received the invitation letter before the first meeting, informing us about the project that was coming here, and then they got in touch via phone to see if they had received the letter. In addition, they wanted to see if the company was interested in joining the project. That's when all the registration started (ENT1)". |
| Identify business opportunities | "( . . . )We participated a lot with ACI. This was discussed within the ICA. ACI is one of the mobilizers of the CE project there, together with FIEMG, presenting these companies. Did you understand? I was very young. It was something like that, identifying what could be created in partnership. It was something that needed to be worked on. (ENT10)". |
| Give credibility to the project | "( . . . ) we know the seriousness of their work (ACI), the proposal they always bring to help companies in general. Mainly in the Sete Lagoas region. So, we know that they are very active, they are trying to bring benefits and proposals that bring us back to the business community here in Sete Lagoas (ENT9)". |
| Mediate relationships between actors | "I characterize the participation of FIEMG as extremely important, mainly as a mediator. To bring companies to work together. To help us define a common ground (ENT6)". |
| Follow the actions | "( . . . ) Until we consolidated the closing of the logistics of wooden packaging, FIEMG acted a lot. If I'm not mistaken, we had a more robust presence from FIEMG for about eighteen months (ENT 6)". |
| Facilitate communication and provide technical expertise | "FIEMG is paramount. They have technical training and contacts with companies throughout the state. That way, they can condense it all. And the ACI plays the same role within the entire state. All companies are directed to it, and it also manages to reduce the information in a single place. This makes the search easier for all companies (ENT7)". |

However, the project itself points to the impossibility of maintaining continuous actions by FIEMG, as there are 11 regional projects distributed throughout the State with different fronts of action. He understands that:

"( . . . ) Even FIEMG is not positioned to be an actor present in each of its micro-regions. So that's why we understand that the industry trade association or the union when it's an industrial centre with the greatest presence of unions, are the major actors in these frequent mobilizations (ENT2)".

In this sense, it would be interesting to establish a local governance structure made up of entrepreneurs, representatives of the local government, local entities and civil society to continue the project. A local actor, the owner of one of the companies participating in

the project, stood out as a possible champion for this IS network under construction in Sete Lagoas.

### 4.2.3. Category 3–Barriers

In the category Barriers to IS implementation in the Sete Lagoas DI, the strongest subcategory is the absence of government actions at the federal, State and municipal levels. The purpose is to promote an environment of dialogue and collaboration with the private sector to think of ways and solutions related to more sustainable industrial practices. In Table 12, the interviewees indicated their expectations regarding the role of the State.

**Table 12.** Subcategory-Absence of Government Actions.

| Role of the State | Coded Excerpt |
| --- | --- |
| Joint actions to propose solutions | "( . . . )We do not have the participation of any competent body assisting us in the solution. I think that the State should encourage participation, a more responsible action. And we don't see that happening. And what is lacking is government participation itself. It lacks structuring (ENT10)".<br>"( . . . ) I think it's even a matter of having more significant support because we would be able to generate many solutions (ENT13)". |
| Support for small businesses | "( . . . )The government itself, I think, is very silent on this. Much more could be done, and more extensive work could be done, especially with small companies. Big companies have a whole structure and full support, but small companies leave something to be desired (ENT3)". |
| Research incentive | "( . . . )We need research. We need resources to continue with these initiatives. So at the federal level, either through the CNI or the federal government. I don't see any genuine initiative to encourage (ENT2)". |
| Engagement in environmental guidelines | "( . . . )Nothing effective about the disposal of materials. Some obligation dictates that we should discard so much, enjoy so much, nothing in this sense as much as there are a series of agreements from Paris. A series of deals are made about the climate, but Brazil is still not very engaged. (ENT4)". |

Another perceived barrier concerns the unavailability of time for the actors involved to dedicate themselves to the project's actions, such as participation in meetings, negotiation rounds, and monitoring of visits. FIEMG realized that the travel time to the ACI headquarters to participate in the discussions could discourage the entrepreneur or company director from getting involved. ENT2 corroborates this finding when reporting the day-to-day rush:

"( . . . )First, companies don't have the time to dedicate to this program. So if there is no one there, in this case, it was FIEMG, but if there was another entity to carry on these mobilizations, it could be the State government or the ACI itself. Anyway, some entity that these industries recognize and that would be the focal point and in this constant to promote these actions without this, I guarantee you that it will not happen ( . . . ) for people to travel from the company to go to the ACI or to go there to the headquarters from FIEMG to attend meetings for the company is a waste of time (ENT2)".

On the barriers encountered in establishing IS practices, the dilemma between the cost of destination versus the cost of investment is highlighted. ENT4 reports your company's waste treatment cost: "To give you an idea, our material treatment cost does not reach 0.1% of the billing. It is deficient, and nobody is interested. But it matters to me because I'm simply taking the material, throwing material away, I throw material in the trash without any reuse (ENT4)".

Faced with this low destination cost, companies do not feel motivated to invest in new practices and technologies and get involved in new arrangements. Finally, although the

project had good adhesion from companies to adopt IS practices, the interviewees pointed out that there was no feedback from the participating companies about the results achieved, new negotiation rounds, or monitoring of actions. This discontinuity of activities was identified as a barrier to the project's progress.

"( . . . )There was an initial follow-up, someone doing an intermediate job, but it seems that it only lasted a short time. There was no great insistence in this sense, you know? I think that if we had, we could suddenly have found some partners not only for the company but for other partners there to develop something (ENT16)".

## 5. Discussion

Understanding the context in which the IS project is inserted, and the history of existing relationships allows us to clarify some points related to the behaviour of organisations in the face of demands for change. When it comes to exogenous elements, it is clear that laws and sanctions are the ones that most determine organisational action. However, this pressure is still focused on the traditional and unidirectional model of production processes in which efforts are aimed at a better final waste destination.

The National Solid Waste Policy (PNRS in Portuguese) [91] is a modern instrument that indirectly encompasses IS and CE through devices that encourage the improvement of production processes, the reuse of solid waste, and the use of energy. So far, developments in plans, programs and actions have not been seen that promote significant changes in waste management in Brazil [21,88,92].

Despite the relevance of legal compliance as an inducer of new organisational practices, a movement of voluntary adherence to norms and procedures is also perceived in companies in the district of Sete Lagoas, especially in medium and large companies. This behaviour is a worldwide trend. The number of ISO 14001-certified organisations in Europe has also been growing steadily [93].

The importance of this new organizational culture was explicit in the study results when respondents spoke about strategic alignment. The fact that some companies in the industrial district are part of larger groups means that the focal company's culture spreads to subordinates. This factor was also observed in the IS experience in Tanzania. The focal company's culture has promoted continuous improvement in the subsidiary processes [47].

In this sense, this change movement has occurred in response not only to environmental forces but also to purposeful and voluntary actions by organisations. Establishing a cooperative environment between companies and government to achieve better environmental results is one of the characteristics pointed out for the success of Kalundborg [30,37,94].

In the context of new practices, demands and technologies, the IS presents itself as a strategy compatible with these concerns. However, for it to occur, some elements proved to be relevant in the study. The economic viability was identified as a critical factor for establishing the IS among the companies of the industrial district in Sete Lagoas, which is in line with the results presented by [31] in cases implemented within the scope of the NISP. For them, the economic viability of the exchanges and solid business opportunities are crucial to the success of the transactions.

Contrary to what was perceived in cases such as Kalundborg and Ulsan, where the community was an active agent to pressure companies to adopt more sustainable practices, resulting in the adoption of IS [55]. This action and community engagement were not noticed in the Sete Lagoas DI. The interface with the community is restricted to social activities such as the revitalization of squares, training courses, and food donations.

Regarding interactions between companies, it was possible to perceive that there was already an interaction before the project between some industrial district companies, especially among companies in the automotive sector, through workshops, lectures and visits. With the project's initiative, this interaction was intensified, including between companies from different sectors. This interaction between companies from the same segment and between companies from various sectors also contributed to identifying synergy opportunities in Kwinana [95]. This exchange of experiences and information

broadens the understanding of the inputs and outputs of other companies, which facilitates the establishment of flows of resources and allows an excellent approximation between these actors.

Similar to what happened in Ulsan [96], in Mexico [11] and in the cases implemented by the NISP, it was found that the CE pilot project increased the institutional capacity of the region to develop IS. After all, progress was made in sharing new knowledge, promoting more significant interaction between organisations and identifying business opportunities for companies.

In order to resolve this issue, several IS studies [46,71,97] point to the importance of the champion or an entity that is responsible for coordinating actions. Throughout the project, ACI and FIEMG were identified as this figure capable of mobilizing and sensitizing actors for being serious entities, with purpose, with trained technical personnel and above all, for inspiring confidence. Therefore, for the maintenance of actions, engagement of new actors and constancy of interactions. It is necessary to establish a local governance structure in which roles are assigned to the actors involved, especially someone who assumes this coordinating role in the industrial district of Sete Lagoas.

Our findings show that public-private partnership is essential for the IS process's cooperative and coordinative facets. The coordination mechanisms developed by FIEMG proactively support the learning process and can collectively help organisations to overcome implementation barriers.

In the analyzed case study, SMEs were also part of the project. They have a significant impact on the Brazilian economy and employment rates. Its awareness of the IS practices and the possibilities for interaction that emerged are necessary to build and encourage a sense of community among the companies. It occurs since any CE project success only will be achieved through collaborative relationships, as reported by Rincón-Moreno et al. (2020) [14].When comparing the results found in the case study of the Sete Lagoas industrial district to the theoretical-analytical model proposed by [55], it appears that the continuity of the project will be conditioned to the improvement of some elements. Among them, we highlight the collective engagement of industrial actors through collaborative environments such as councils and committees to solve problems together; partnerships with teaching and research institutions for the development of new technologies, products and processes; strengthening of information management in order to make relationships more transparent and reliable; fostering the heterogeneity of industrial actors by attracting new companies to the Sete Lagoas industrial district or encouraging the creation of new companies; strengthening and creating spaces that promote the participation of the local community in decisions; review of the current fiscal and tax policy combined with the implementation of incentives and financial support for companies that adopt circular measures and, finally, strengthening the collaborative culture and awareness of environmental issues. Figure 4 summarizes the categories and subcategories that emerged from the analysis of the survey data.



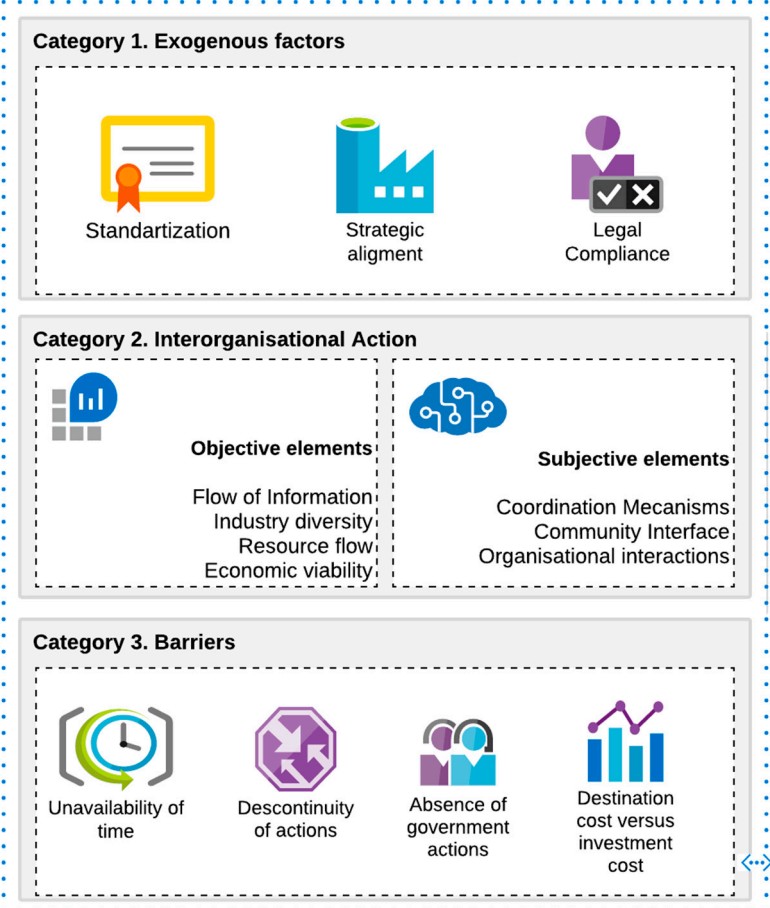

**Figure 4.** Main elements of the Industrial Symbiosis process in the Industrial District of Sete Lagoas.

## 6. Conclusions

As the analysis of the transition process to Industrial Symbiosis (IS) in the industrial district of Sete Lagoas was carried out, units of meaning were discovered, and from them categories and subcategories emerged. An underlying structure to the IS implementation process was perceived, which includes: the exogenous elements that unfold in laws, norms and shared understandings; the objective elements of organisational action, which are resource flows, industrial diversity, economic viability and information flows; the subjective elements of the relationships that were divided into the interface with the community, organisational interactions and the coordination role; and finally, the barriers that are: absence of governmental actions, unavailability of time, cost of allocation versus cost of investment and discontinuity of actions.

Although the Sete Lagoas CE pilot project outcomes could have been more impressive, the findings of this paper have theoretical, practical, and social implications. First, this study contributes to the development of IS literature, gathering and synthetizing the main knowledge on IS, which can be helpful to researchers interested in this topic. The results allow a greater understanding of the variables involved in the evolutionary trajectory of Industrial Symbiosis. Concerning social implications, applying the IAD framework and, consequently, all the constructs related to Ostrom's work to IS process allowed us to get a micro and meso-level understanding from the point of view of all stakeholders involved willing to collaborate. The actors use cooperative and coordinating collaboration mechanisms to build up knowledge on IS and collectively overcome implementation barriers on the individual, organizational, and institutional levels.

In terms of management implications, this research offers insights into IS adoption in an emerging country such as Brazil, which can be replicated in other South America countries. The IS implementation process brought awareness about such issues as the reuse

of by-products, cascading flows, and co-processing, which go beyond knowledge about sustainable waste management. As the leading industry stakeholders became aware of the need to balance economic, social, and environmental dimensions, there is an expectation to adopt new models such as IS. At the same time, this has clear implications for society regarding environmental protection, job creation, and innovation in industrial sectors, which can also generate more economic return to the local government and community.

From a practical point of view, the exploratory and descriptive analysis of the Sete Lagoas CE pilot project can serve as a model for other industrial districts that are in the same situation and have the same characteristics. Therefore, it can support the planning of local government actions to improve the current project. It can foster policies aimed at practices that focus on the circularity of resources. In addition, it allows a reflection on the scarcity of natural resources and the recovery of waste as an economic good. These results can be helpful for researchers studying this topic and managers in Brazil and other emerging countries in Latin America, as well as, policymakers involved in public policies aimed to enable the transition to a circular and more sustainable model.

Despite of its contributions, this article has some limitations to be pointed out. The main limitation of the study is related to the methodological choices made. As for the data collection technique used, the interviews, there is a propensity for bias in the responses. In this sense, efforts were made to create instruments and conduct the interviews to allow for minimal bias. Also, regarding the methodology, we cannot make generalizations because it is a case study. We also did not use any organizational theory as a lens of analysis of results, just using the framework from Ostrom [98] as a basis, because it fits better the purpose of the paper.As a suggestion for future studies, it is suggested to deepen the study from a quantitative analysis of the flows established in the industrial district of Sete Lagoas. Methods such as life cycle assessment can help calculate these exchanges' financial and environmental impacts to encourage other companies' participation. We suggest that future studies address the phenomenon under the analysis of stakeholders or institutional theories to advance. It can be analysed how the stakeholder's relationship can enable or difficult the interaction and collaboration between companies in an IS arrangement. The Institutional theory can also be used to understand how the environment can affect the behaviour of the companies acting in these arrangements.

**Author Contributions:** Conceptualization, writing and methodology, E.F.; formal analysis, E.F. and C.B.; methodology supervision, C.B.; supervision, A.C.-P.; writing review and editing, J.A.C.S. and P.G. All authors have read and agreed to the published version of the manuscript.

**Funding:** This research received no external funding.

**Institutional Review Board Statement:** Ethical review and approval were waived for this study due to involving just the participants' opinions and not involving any experiment with humans or animals, did not involve any vulnerable population and sensitive topic.

**Informed Consent Statement:** Informed consent was obtained from all subjects involved in the study.

**Data Availability Statement:** Data sharing does not apply to this article as no datasets were generated or analyzed during the current study.

**Acknowledgments:** The authors are grateful to FIEMG and ACI for their support. We are very grateful for the comments and suggestions offered by the two anonymous reviewers.

**Conflicts of Interest:** The authors declare no conflict of interest.

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
