# Peer review of "Brazilian Circular Economy Pilot Project: Integrating Local Stakeholders’ Perception and Social Context in Industrial Symbiosis Analyses"

_sustainability, doi:10.3390/su15043395_

Round 1

Reviewer 1 Report

The objective of the article “Brazilian Circular Economy Pilot Project: An analysis of the behaviour of organisations towards industrial symbiosis” is to explore the organizations' behaviour in the process of implementing industrial symbiosis. For this purpose, a case study of a pilot project in Brazil was conducted from the stakeholders' point of view. The authors used an in-depth interview method and content analysis technique.

Based on the analysis of the transition process to industrial symbioses, it was possible to identify the elements that constitute the structure of the implementation process, i.e, exogenous elements, objective and subjective elements, as well as barriers.

The study conducted in the Brazilian context has theoretical implications by filling a research gap with data on developing countries regarding the process of industrial symbiosis. At the same time, the study can serve as a model for other industrial districts, and support local government action planning.

This is a well-written account of the industrial symbiosis process in the Brazilian industrial district. Overall, there are several promising aspects in the manuscript.

Below are my comments and suggestions for the Authors:

Regarding the interviews conducted, did you not consider the impact of the particular interviewee's position and work experience (years of work)? This is significant because research shows that employees at different levels of a company often have very different views. Regarding the available entities, have you not considered interviewing by employee position, e.g., analyst group, manager group, CEOs group? What are the limitations of the study stemming from the data collection methods? – should be described more clearly in Materials and Methods section.

Please also consider minor technical issues:

- When the abbreviation IS first appears in the abstract - please explain it i.e., its meaning in brackets or use whole words in the abstract.

- In line 69, or 100 - the abbreviation PPE - where is the meaning indicated?

- In line 90 there is no need to translate the abbreviation another time since "IS" is already explained in the earlier paragraph, similarly the abbreviation NISP is translated again in the Results section.

- Several times there is a mistake in writing the abbreviation industrial symbiosis e.g., lines 64, 172

- In line 56 - Ecoindustrial parks (EPI) - is this a good abbreviation? Wasn't it supposed to be EIP? Then in line 176 the abbreviation "EIP" is used; in addition, a better notation is “eco-industrial parks”

- Appropriate “CO2” notation (with a small subscript - CO2)

 - Quoting the paper as in lines 168 - 170, citing the authors should be written "According to Boom Cárcamo et al. [41]" and "Faria et al. [42] showed that" - because as it is now, the subject is missing.

 - What does the abbreviation EC mean? The abbreviation CE (circular economy) is explained, while EC what is the meaning? The same thing? - please review all abbreviations and acronyms used

 - Sometimes a capital letter is used unnecessarily after a colon or in the middle of a sentence, e.g., in line 586

Reviewer 2 Report

I would like to thank you for the opportunity to review the manuscript entitled: Brazilian Circular Economy Pilot Project: An analysis of the behavior of organizations towards industrial symbiosis.

 Overall the article is well written, but could be improved. Here are my considerations, I hope that with them it will be possible to improve the paper.

Title: After possible adjustments indicated below, assess whether the title can be improved to better reflect the research objectives, as well as demonstrate originality, if any.

 Abstract: What makes research interesting for academia, for professionals, for governments? What is the originality of the research? This needs to be made clear in the summary.

The exogenous elements mentioned in the summary I think refer to the Stakeholder Theory. Make it clear. And, if they haven't talked about it in the text (literature review), it would be interesting to do so.

Introduction: I suggest that the authors include what are the objectives of the work, what are the issues and problems. In addition, it is necessary to make clear the importance of the work, what research gap do the authors want to fill? Justify this by citing good and recent references. Why they chose Minas Gerais – Brazil (these companies), this needs to be clear.

In the introduction I noticed that from reference number 2 the authors already jump to reference 4...where reference 3 is. Please correct errors of this type here and throughout the work.

In 2.1 It would be interesting to bring a recent definition of Industrial Symbiosis.

In 2.1, starting from paragraph 3 the authors write: Although the initial development of IE was based almost exclusively on technology based arguments [1], there has been a movement in recent years that values the contributions of the social sciences to the field. This new approach seeks to understand to what extent material and energy flows are shaped by the social context in which they are in-105 serted [24]. In this sense, studies have been conducted emphasizing the interactions of the 106 industrial system with the environment. Contextual elements, organizational structure, 107 interaction patterns, managers' beliefs and how governments try to influence the behavior of organizations began to compose the analyses. 109

The UK has several IS initiatives fostered from the NISP. In the first studies, [25] 110 highlighted the relevance of some factors, such as the nature of the companies' operations 111 and industrial history, peer pressure and the coordination mechanism. In more recent analyses, the factors geographic proximity and social interactions were questioned by [26] and [23]….see that the authors mention that there is a recent movement, recent research, but the references they use to talk about this “recent” movement already have more than 10 years of publication. Review this.

In general, the literature review carried out needs improvements (it is old and not complete). For example, if you are mentioning Industrial Symbiosis, you should make a brief evolution of the theme, define the theme, talk about the importance of the theme, its main practices, benefits, etc.

Regarding industrial symbiosis in Brazil, are you sure that in the literature we only have those projects? In addition to projects, there are no recently published articles that deal with the topic??? It is necessary to improve the literature review on the subject.

Materials and methods: unclear. It is necessary to justify each choice, each procedure and cite the reference that supports them. It is necessary to describe in more detail the “step by step” from choosing the type of research, selecting the cases, creating the data collection instrument, planning and analyzing the data, who was interviewed, the duration of the interviews, etc. For previous cases where there is already written text, I suggest following a logical sequence, perhaps dividing this section into subsections, for example: methodological choices, case selection, data collection instrument, Data collection, data analysis.

On line 239 the authors say: “The framework is divided into three broad categories: exogenous 240 factors, inter-organisational action and barriers to implementation. The subcategories help 241 explain the main categories of analysis from the grouped recording units.” Explain better how these categories were created, based on which theoretical framework? Cite the source!!! If there is a theoretical framework, it must be in the previous section of the literature review.

Results: In 4.1 there are some paragraphs with statements that I couldn't identify the source. Review giving credits from where you took the information.

Much of the text in section 4.1 I believe is not result text. Please review this! I think that the most appropriate place is in the literature review (it would even be interesting to have a section with that name).

 In 4.3 it needs to be better explained (as already mentioned) in the literature review where these classifications/categories and subcategories came from. Cite the references. Make it clear in the literature review why you chose this classification. Explain them.

In general, the article has a lot of results, but I believe that they can be better used. A larger and better comparison with recent literature is needed. Where the work is similar, where the work is different from current relevant literature (discussions).

As for the conclusions, a better explanation of the importance of the work is needed. Discuss more about the theoretical and practical implications. What are the implications for managers, professionals, governments. How is this research different from others already carried out? It is necessary to understand the originality of the research, which is still not clear.

 References: I noticed that several references that deal with the main theme of the article are old... they are 10 or more than 10 years old. Update these references by always adding some new ones to them.

Finally, I would like to see in the paper which organizational theories the authors were based on (motivated) to develop the work. Justify!

Round 2

Reviewer 2 Report

Dear Editor and Authors.

Dear Editor and Authors.

Thanks again for the opportunity to review the paper. I can see that the authors have worked hard on the requested points and I understand that the article is now much better. I just suggest one last look at spelling and formatting according to journal guidelines. Congratulations! Success!